# Efficacy of Cytoreductive Surgery (CRS) + HIPEC in Gastric Cancer with Peritoneal Metastasis: Systematic Review and Meta-Analysis

**DOI:** 10.3390/cancers16101929

**Published:** 2024-05-18

**Authors:** Lodovica Langellotti, Claudio Fiorillo, Giorgio D’Annibale, Edoardo Panza, Fabio Pacelli, Sergio Alfieri, Andrea Di Giorgio, Francesco Santullo

**Affiliations:** 1General Surgery Department, Catholic University of the Sacred Hearth, 00168 Rome, Italy; lodovicalangellotti@yahoo.it (L.L.); dannibalegiorgio@gmail.com (G.D.); edoardo.panza01@icatt.it (E.P.); fabio.pacelli@policlinicogemelli.it (F.P.); sergio.alfieri@policlinicogemelli.it (S.A.); 2Department of Digestive Surgery, Fondazione Policlinico Universitario Agostino Gemelli IRCCS, Catholic University of the Sacred Hearth, 00168 Rome, Italy; 3Department of Peritoneum and Retroperitoneum Surgery, Fondazione Policlinico Universitario Agostino Gemelli IRCCS, Catholic University of the Sacred Hearth, 00168 Rome, Italy; andrea.digiorgio@policlinicogemelli.it (A.D.G.); francesco.santullo@policlinicogemelli.it (F.S.)

**Keywords:** gastric cancer, peritoneal metastasis, cytoreductive surgery, HIPEC

## Abstract

**Simple Summary:**

Peritoneal disease in gastric cancer has a poor prognosis, with a median survival of 3–6 months and a 5-year survival rate of 0%. Despite multiple advancements in therapeutics, the National Comprehensive Cancer Network (NCCN) guidelines recommend systemic chemotherapy or best supportive care for GC with peritoneal dissemination. According to several studies, CRS + HIPEC could provide survival advantages in gastric cancer peritoneal metastasis compared to pSC.

**Abstract:**

Background: Peritoneal carcinomatosis is one of deadliest metastatic patterns of gastric cancer, being associated with a median overall survival (OS) of 4 months. Up to now, palliative systemic chemotherapy (pSC) has been the only recommended treatment. The aim of this study is to evaluate a potential survival benefit after CRS + HIPEC compared to pSC. Methods: A systematic review was conducted according to the PRISMA guidelines in March 2024. Manuscripts reporting patients with peritoneal carcinomatosis from gastric cancer treated with CRS + HIPEC were included. A meta-analysis was performed, comparing the survival results between the CRS + HIPEC and pSC groups, and the primary outcome was the comparison in terms of OS. We performed random-effects meta-analysis of odds ratios (ORs). We assessed heterogeneity using the Q2 statistic. Results: Out of the 24 papers included, 1369 patients underwent CRS + HIPEC, with a median OS range of 9.8–28.2 months; and 103 patients underwent pSC, with a median OS range of 4.9–8 months. CRS + HIPEC was associated with significantly increased survival compared to palliative systemic chemotherapy (−1.8954 (95% CI: −2.5761 to −1.2146; *p* < 0.001). Conclusions: CRS + HIPEC could provide survival advantages in gastric cancer peritoneal metastasis compared to pSC.

## 1. Introduction

Gastric cancer is the fourth leading cause of cancer death in the world, with a global 5-year survival rate of 25% [1]. Peritoneal carcinomatosis is one of the most common metastatic patterns that develops in 20–30% of patients with gastric cancer, and it is associated with a poor prognosis, with a median overall survival of 4–6 months without treatment, and no long-term survival [2]. According to the current guidelines, systemic palliative chemotherapy (pSC) and best supportive care are the only recommended treatments [3]. Unfortunately, chemotherapy alone brings unsatisfactory results, mainly due to the inadequate ability of chemotherapy drugs to penetrate the blood–peritoneal barrier and to act on the peritoneal implants [4,5].

Since the 1990s, to overcome the poor results of pSC, the concepts of “cytoreduction” and intraperitoneal chemo-hyperthermia (HIPEC) were proposed. Cytoreductive surgery requires the surgical removal of all macroscopically visible disease; meanwhile, HIPEC consists of the administration of a chemotherapeutic drug directly into the abdominal cavity, in contact with the peritoneal metastases, better ensuring its penetration into the tissues by taking advantage of the induced hyperthermia.

Cytoreductive surgery in association with HIPEC is a treatment modality considered standard of care for some tumors that are metastatic to the peritoneum [6,7], but its role in the treatment of peritoneal carcinomatosis from gastric cancer is still controversial.

The efficacy of cytoreductive surgery and HIPEC in the treatment of gastric cancer should be confirmed in randomized clinical trials conducted with a high number of patients. Therefore, this treatment should still be considered experimental and can be proposed only in the context of clinical studies in high-volume referral centers for peritoneal carcinomatosis treatment.

The purpose of this systematic review and meta-analysis is to confirm the safety, the efficacy, and the superiority of CRS + HIPEC compared to palliative systemic chemotherapy (pCS) in patients with stage IV gastric cancer exhibiting synchronous peritoneal metastasis, as well as to identify if there are any specific characteristic for an “ideal candidate” for this treatment.

## 2. Materials and Methods

### 2.1. Data Sources, Search Strategy, and Selection Criteria

A systematic review of published papers was conducted according to the Preferred Reporting Items for Systematic Reviews and Meta-Analyses (PRISMA) guidelines in March 2024 [Figure 1].

The systematic research was conducted on PubMed and Scopus, searching for papers reporting cases of patients with stage IV gastric cancer with exclusive and synchronous peritoneal metastasis, treated with cytoreductive surgery plus HIPEC.

The search terms “gastric cancer”, “peritoneal metastasis” and “cytoreductive surgery” were adapted for each database. The final search was as follows: “Stomach Neoplasms” [MeSH Terms] OR “stomach neoplasm” [tw] OR “stomach cancer” [tw] OR “stomach carcinoma” [tw] OR “stomach tumor” [tw] OR “stomach tumour” [tw] OR “stomach malignancy” [tw] OR “pancreatic ductal carcinoma” [tw] OR “gastric neoplasm” [tw] OR “gastric cancer” [tw] OR “gastric carcinoma” [tw] OR “gastric tumor” [tw] OR “gastric tumour” [tw] OR “gastric malignancy” [tw] AND “Peritoneal Neoplasm” [MeSH Terms] OR “peritoneal surface malignancy” [All Fields] OR “peritoneal carcinomatosis” [All Fields] OR “peritoneal synchronous metastasis” [All Fields] OR “peritoneal carcinomatosis” [All Fields] OR “peritoneal carcinosis” [All Fields] OR “peritoneal metastasis” [All Fields] OR “synchronous metastasis” [All Fields] OR “carcinomatosis” [All Fields] OR “carcinosis” [All Fields] AND “cytoreductive surgery” [MeSH Terms] OR “cytoreduction” [All Fields] OR “peritonectomy” [All Fields] OR “HIPEC” [All Fields] OR “surgery” [All Fields] OR “surgical resection” [All Fields] OR “gastrectomy” [All Fields].

The references of the included articles were also manually searched, and further articles were included if appropriate. The selection criteria included “English” language, human studies, clinical trials, and observational and comparative studies. Duplicate references were semi-automatically removed using the Rayyan platform (https://www.rayyan.ai/, accessed on 15 March 2024). Case reports were excluded. The articles were uploaded into the Systematic Review Accelerator (www.sr-accelerator.com, accessed on 15 March 2024), which is a web-based screening tool. The abstracts and titles were screened independently on the Systematic Review Accelerator by two reviewers (L.L. and F.S.). Each paper retrieved was assessed for inclusion or exclusion by two authors through revision of titles and abstracts (L.L and F.S), and any issues or disagreement was resolved by consensus between the authors. This systematic review and meta-analysis was registered on PROSPERO (registration number: CRD42023495346).

### 2.2. Study Risk-of-Bias Assessment

Randomized controlled trials (RCTs) and non-randomized clinical trials (n-RCTs) were evaluated using, respectively, the Robin 2.0 tool and the Robin-I tool, whereas non-randomized retrospective cohort studies were scored using the Newcastle–Ottawa scale [8,9,10,11,12,13,14,15,16,17,18,19,20,21,22,23,24,25,26,27,28,29,30,31].

### 2.3. Inclusion and Exclusion Criteria

Adults (>18 years), both male and female, with gastric cancer and synchronous peritoneal carcinomatosis who underwent cytoreductive surgery and HIPEC were included. Patients with gastric cancer and other sites of metastasis were excluded.

### 2.4. Outcomes of Interest

The primary outcome of interest was the median overall survival of patients with gastric cancer and synchronous peritoneal carcinomatosis who underwent cytoreductive surgery and HIPEC. The secondary outcomes were 1-, 2-, and 5-year survival rates, disease-free survival, and grade III-IV complication rate.

### 2.5. Statistics

The meta-analysis was conducted using Jamovi software (version 2.4.11.0). The analysis was carried out using the log odds ratio (OR) as the outcome measure. The OR and 95% confidence intervals (95% CI) were calculated to estimate the association between binary factors and AL. A fixed-effects model was fitted to the data. Furthermore, the Q-test for heterogeneity (Cochran 1954) and the I^2^ statistic were reported. Studentized residuals and Cook’s distances were used to examine whether studies may be outliers and/or influential in the context of the model. The studies with a studentized residual larger than the 100 × (1 − 0.05/(2 × k))th percentile of a standard normal distribution were considered potential outliers (i.e., using a Bonferroni correction with two-sided alpha = 0.05 for k studies included in the meta-analysis). Studies with a Cook’s distance larger than the median plus six times the interquartile range of the Cook’s distances were considered to be influential. Finally, the rank correlation test and the regression test, using the standard error of the observed outcomes as a predictor, were used to check for funnel plot asymmetry.

## 3. Results

### 3.1. Studies Selection and Patient’s Characteristics

Using the search strategies described above [Figure 1], 4256 papers were identified. After removing 1167 duplicates, 3089 were screened. Out of 3089 papers, 2949 were removed after reading the abstract. Exclusion criteria were as follows: (1) case report and case series; (2) full text not available; (3) papers not in English; (4) years of publication before 2011; (5) systematic reviews and meta-analyses; (6) narrative reviews, editorials, and conference abstracts; (7) irrelevant topic. Then, out of 140 papers, after accurate full-text reading, 116 were excluded because of wrong population, outcome, or study design. Finally, 24 papers met the inclusion criteria and were included. Table 1 shows the 24 papers [8,9,10,11,12,13,14,15,16,17,18,19,20,21,22,23,24,25,26,27,28,29,30,31] analyzed, which including 1369 patients with gastric cancer and peritoneal carcinomatosis who underwent cytoreductive surgery plus hyperthermic intraoperative chemotherapy.

### 3.2. Studies’ Risk-of-Bias Assessment

There were two RCTs [8,29], three non-randomized clinical trials [14,23,28], and 19 non-randomized retrospective cohort studies [9,10,11,12,13,15,16,17,18,19,20,21,22,24,25,26,27,30,31]. All non-randomized studies scored 7 or more on the Newcastle–Ottawa scale, and both the n-RCTs and RCTs had a low risk of bias according to Robin-I and Robin 2.0. All studies were therefore deemed to be good quality studies [Appendix B].

### 3.3. Surgical Treatment

Cytoreductive surgery was performed, starting with the resection of the primary tumor and with total or subtotal gastrectomy, associated with D2 lymphadenectomy. Selective peritonectomy and multiple organ resections were performed, depending on disease extension, to achieve CC-0 resection. HIPEC was performed using either the open or closed technique. Several HIPEC regimens were employed, including cisplatin, mitomycin, paclitaxel, doxorubicin, oxaliplatin, 5-fluorouracile, docetaxel. The temperature range was 40–43.5 °C, and the duration range was 30–120 min. The neoadjuvant therapies and regimens are summarized in Appendix A.

### 3.4. Survival, Recurrence, and Surgical Outcomes of CRS + HIPEC

Out of the twenty-four studies included, twenty-one reported the median OS of patients who underwent CRS + HIPEC, with a range of 9.8–28.2 months [Figure 2].

The median OS of the studies included was 17 months. Respectively, the 1-, 3-, 5-year rate survival ranges were 40.9–96%, 5.9–78%, and 0–55%. The median DFS range was 7.1–12 months. [Table 2]

Thirteen papers reported the 30-day mortality after CRS + HIPEC: out of 837 patients, 25 died within 30 days from surgery (3%). [Table 2]

Sixteen studies reported the complication rate incidence (grade III–IV Clavien Dindo): out of 1034 patients, 249 experienced a major post-operative complication (24%). [Table 2]

### 3.5. CRS + HIPEC versus Palliative Chemotherapy

Four studies comparing the survival outcomes of CRS + HIPEC versus palliative systemic chemotherapy were included for the meta-analysis. CRS + HIPEC was associated with significantly increased survival compared to palliative systemic chemotherapy. The pooled result is showed in Figure 3 (−1.8954 (95% CI: −2.5761 to −1.2146; *p* < 0.001)).

The Q-test for heterogeneity was not significant, but some heterogeneity may still be present in the true outcomes (Q(3) = 7.0140, *p* = 0.0715, I2 = 57.2282%). An examination of the studentized residuals revealed that one study [26] had a value larger than ± 2.4977 and may be a potential outlier in the context of this model. According to the Cook’s distances values, none of the studies could be overly influential. The regression test indicated funnel plot asymmetry (*p* = 0.0182), but the rank correlation test did not (*p* = 0.3333) [Figure 3].

## 4. Discussion

Gastric cancer (GC) is the fifth most common cause of malignancy and the fourth leading cause of cancer-related mortality worldwide, with a 5-year survival rate of 25%. Up to 40% of patients with gastric cancer have synchronous peritoneal metastasis at the moment of diagnosis [1,2,3,4,5].

Peritoneal disease has a poor prognosis, with a median survival of 3–6 months and a 5-year survival rate of 0%. Despite multiple advancements in therapeutics, the Italian Guidelines [3] recommend systemic chemotherapy or best supportive care for GC with peritoneal dissemination, whereas the latest NCCN guidelines suggest considering CRS + HIPEC in well-selected patients [32].

The effect of systemic chemotherapy on peritoneal metastasis is limited, probably because of the peritoneum–plasma barrier, which prevents effective drug delivery from the systemic circulation into the peritoneal cavity. Thus, in the last 30 years, several multimodal therapies have been introduced, including hyperthermic intraoperative chemotherapy (HIPEC) associated with cytoreductive surgery, and pressurized intraperitoneal aerosol chemotherapy (PIPAC) [33].

Based on the current literature, only a limited number of studies demonstrate a survival advantage of cytoreductive surgery (CRS) and hyperthermic intraperitoneal chemotherapy (HIPEC) compared to palliative systemic chemotherapy (PSC), and none of them involves a large sample size [34,35,36,37]. Our paper summarizes the evidence potentially supporting the survival benefits of CRS and HIPEC. To our knowledge, it represents the first systematic review and meta-analysis on this topic to include a substantial population size.

In 1996, Yonemura et al. [38] published the results of the first large clinical trial on the efficacy of HIPEC (mitomycin 30 mg + cisplatin 300 mg+ etoposide 150 mg, 60 min at 42–43 °C) in combination with aggressive cytoreductive surgery, including gastrectomy, extended regional lymphadenectomy, and partial or subtotal peritonectomy in GC patients with peritoneal carcinomatosis. They achieved 1-year survival in 43% of patients and, for the first time, 5-year survival was seen in 11% of the patients.

In a prospective clinical trial, Glehen et al. [39] demonstrated that an aggressive management strategy combining cytoreductive surgery and hyperthermic chemotherapy is effective for patients with gastric cancer and peritoneal carcinomatosis, and this approach may result in long-term survival compared to standard therapies. Moreover, Glehen et al. identified more prognostic risk factors correlated with better survival and, according to their study, reaching a complete cytoreduction seems to be the most important aspect.

Given the observed survival advantages, it is noteworthy that cytoreductive surgery combined with hyperthermic intraperitoneal chemotherapy (HIPEC) constitutes a complex procedure, associated with a mortality rate of 6.5% and a morbidity rate of 28.8% (grade III-IV according to the Clavien–Dindo classification) [40]. Our study corroborated similar rates, with a major complication rate of 24% and a mortality rate of 3%. Considering the elevated occurrence of surgical complications alongside the substantial learning curve inherent in cytoreductive surgery [41], a careful selection of patients should be performed.

To date, the eligibility for cytoreduction hinges on two primary factors: the extent of disease, as assessed by the Peritoneal Cancer Index (PCI), and the potential for achieving complete cytoreduction (CC0).

In this systematic review and meta-analysis, we have reaffirmed the robust correlation between the extent of surgical cytoreduction and survival outcomes. The analysis of nine included studies revealed superior survival rates among patients achieving complete surgical cytoreduction (CCR-0), in contrast to those undergoing incomplete cytoreduction (CCR 1–2).

PCI (Peritoneal Carcinosis Index), used to quantify the extension of pre-operative peritoneal carcinosis, has been reported as an independent prognostic factor in patients with gastric cancer. Our findings corroborate this concept, with eleven of the studies included reporting the correlation between a lower PCI and better OS in the multivariate analysis.

In 2010, Glehen et al. [39] recommended a PCI limit of 12, having observed that no patients with a PCI > 12 survived in their analysis of 159 patients with gastric cancer and peritoneal metastasis (PM). Their study has marked a significant milestone in the guidelines for several years; however, the current trend is moving towards even stricter PCI limits.

Subsequent studies [13,17,20] have reported improved overall survival (OS) in patients with a PCI < 7. Chia et al. [13] found a median OS of 26.4 months for patients with a PCI < 7, compared to 10.9 months for those with a PCI ≥ 7. The Spanish Registry [17], published in 2019, analyzed 88 patients and observed that those with a PCI < 7 had a median OS of 26.1 months (5-year OS of 46.8%), whereas those with a PCI ≥ 7 had a median OS of 18.9 months (5-year OS of 0.0%). Similarly, in 2020, the German Registry [20], involving 235 patients, demonstrated improved OS, with a PCI < 7 (median OS of 18 months for PCI 0–6, 12 months for PCI 7–15, and 5 months for PCI 16–39).

Despite these findings, a precise PCI cut-off associated with better survival outcomes has not been identified yet. Moreover, this numeric score lacks the ability to differentiate the presence of peritoneal metastasis at critical anatomical sites, leading to an inability to distinguish between nodules that are easily removable and those situated in locations that limit resectability, such as widespread nodules in the mesentery and bowel.

Therefore, to assist the decision-making process, it is mandatory to consider other prognostic factors, together with the PCI score and completeness of cytoreduction.

To date, a novel concept, known as oligometastatic disease (OMD), is gaining recognition as a distinct entity from poli-metastatic disease across various cancers, including gastric cancer [42]. Patients displaying minimal peritoneal dissemination are categorized as having OMD, representing an intermediary phase between localized and widespread metastatic conditions. The ongoing RENAISSANCE trial [43] aims to precisely define OMD and explore the potential role of surgical intervention (including excision of the primary tumor and localized removal of metastases), along with perioperative chemotherapy regimens (such as fluorouracil, leucovorin, oxaliplatin, and docetaxel (FLOT), administered over four cycles) in this specific patient population. This study’s outcomes may solidify OMD as a distinct oncological concept, emphasizing the variability in biological characteristics and behaviors observed in gastric cancer. In this context, gaining a comprehensive understanding of the molecular dynamics of OMD and identifying biomarkers are essential for accurately classifying patients and tailoring treatment plans. For patients with biologically confirmed OMD, a more aggressive chemotherapy approach is recommended to achieve an optimal response, allowing cytoreductive surgery and HIPEC.

There are some potential limitations to the present systematic review. The effects estimated in the model are based mainly on retrospective observational studies. They are therefore subjected to biases and confounding factors, which may have influenced our model estimates. The main limitation of the included studies lies in the heterogeneity of the population. Heterogeneity in studies has been reported for a variety of reasons, including differences in the sample population regarding age, sex, BMI, pre-operative PCI, and pre-operative therapies. Another limitation is the different study designs of the included papers (19 observational studies, three non-randomized clinical trials, and two randomized clinical trials); however, only observational studies were included in the meta-analysis. Drawing conclusions based on the moderate effect estimate from the meta-analysis and the certainty of the evidence according to GRADE criteria, CRS + HIPEC probably provides survival advantages in gastric cancer peritoneal metastasis compared to pSC.

## 5. Conclusions

Gastric cancer with synchronous peritoneal metastasis treated with cytoreductive surgery plus HIPEC (CRS + HIPEC) seems to be promising and may have better survival outcomes compared to palliative systemic chemotherapy (pSC), especially in selected categories of patients, with a low pre-operative PCI, and where a complete cytoreduction is technically achievable.

## Figures and Tables

**Figure 1 cancers-16-01929-f001:**
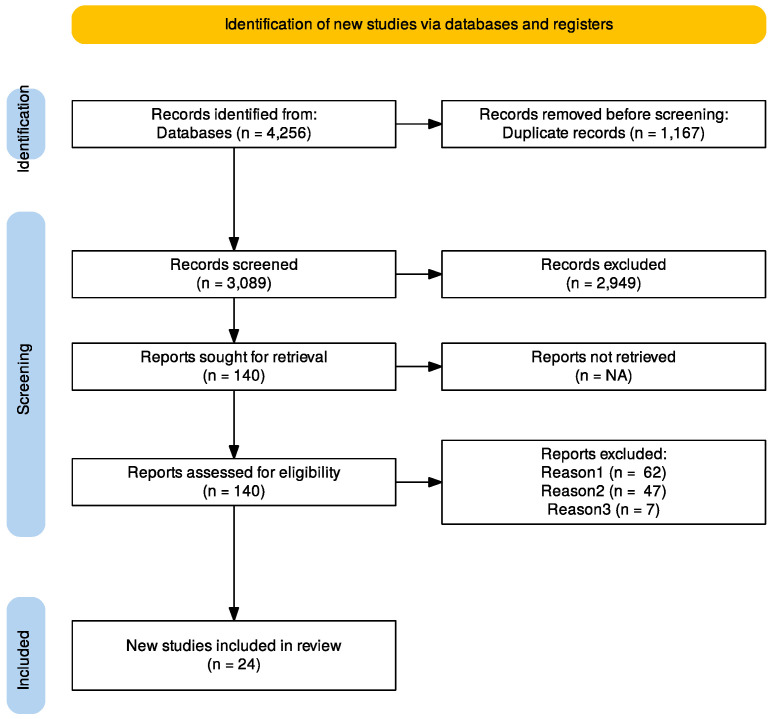
PRISMA 2020 flow diagram for the systematic review. Records identified from PubMed (n = 1166) and Scopus (n = 3090). Records excluded for different population (reason 1), different outcome (reason 2), different study design (reason 3).

**Figure 2 cancers-16-01929-f002:**
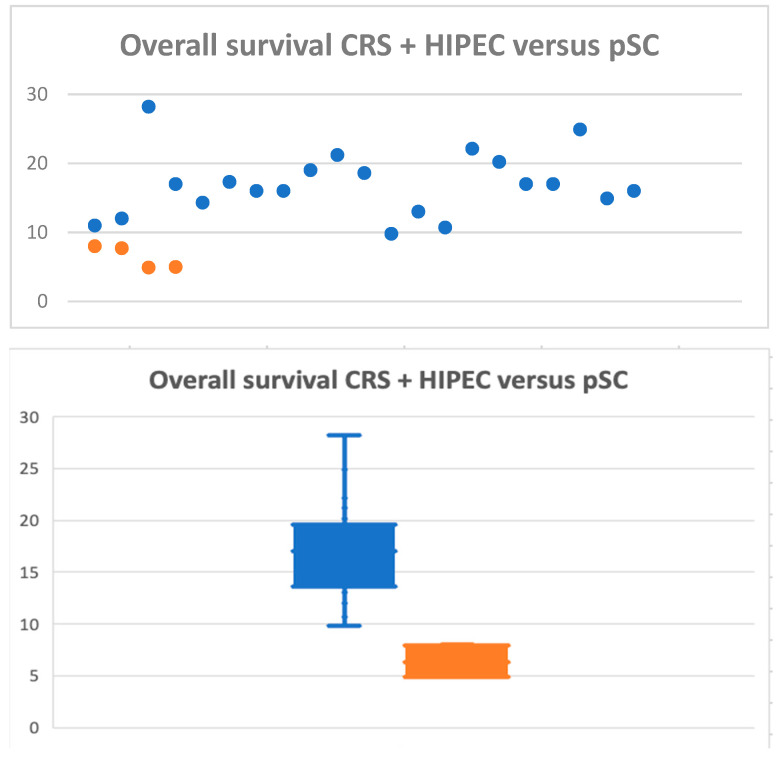
Graphical distribution of median OS of studies included, comparing CRS + HIPEC (blue) and pSC (orange).

**Figure 3 cancers-16-01929-f003:**
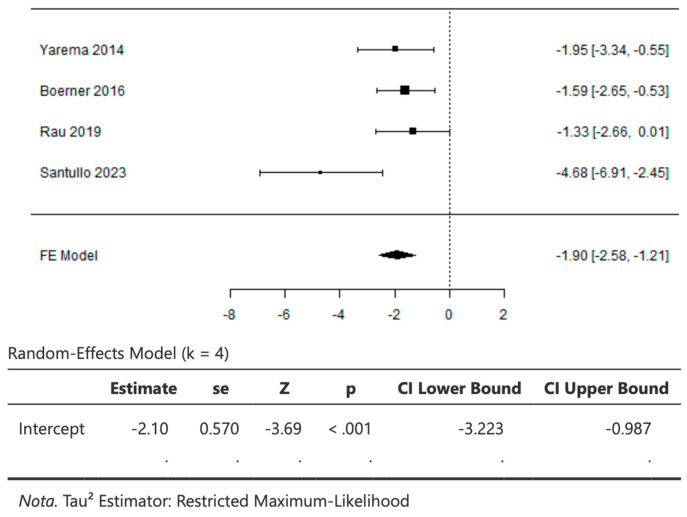
Forrest plot showing survival rates comparing CRS + HIPEC and pCS [9,12,19,26].

**Table 1 cancers-16-01929-t001:** A systematic review of literature: The 24 papers included.

Author, Year	Study Design	N CRS + HIPEC	Median Age	M/F
**Yang, 2011** **[8]**	RCT	34	50	16/18
**Yarema, 2014 [9]**	Retrospective Cohort study	20	NA	10/10
**Wu, 2015** **[10]**	Retrospective Cohort study	11	64	8/3
**Wu H-T, 2016** **[11]**	Retrospective Cohort study	38	NA	NA
**Boerner, 2016** **[12]**	Retrospective Cohort study	50	NA	22/28
**Chia, 2016** **[13]**	Retrospective Cohort study	81	51	37/44
**Topal 2017** **[14]**	Non-randomized prospective trial	32	58	20/12
**Caro, 2018** **[15]**	Retrospective Cohort study	35	53	17/18
**Kim, 2018** **[16]**	Retrospective Cohort study	38	45.8	12/26
**Manzanedo, 2019** **[17]**	Retrospective Cohort study	88	53	43/45
**Bonnot, 2019** **[18]**	Retrospective Cohort study	180	51.13	83/97
**Rau, 2019** **[19]**	Retrospective Cohort study	58	54.6	38/20
**Rau, 2020** **[20]**	Retrospective Cohort study	235	53.4	113/122
**Zhong-He Ji, 2020** **[21]**	Retrospective Cohort study	125	51	59/66
**Rosa, 2021** **[22]**	Retrospective Cohort study	23	52	10/13
**Bagdwell, 2021** **[23]**	Clinical trial	20	58	13/7
**Marano 2021** **[24]**	Retrospective Cohort study	91	58	46/45
**Somashekhar, 2022** **[25]**	Retrospective Cohort study	16	55.5	NA
**Santullo, 2023** **[26]**	Retrospective Cohort study	20	62	8/12
**Buckarma, 2023** **[27]**	Retrospective Cohort study	22	56	17/5
**Green, 2023** **[28]**	Clinical Trial	41	57	26/15
**Rau, 2023** **[29]**	RCT	59	56	29
**Allievi, 2023** **[30]**	Retrospective Cohort study	27	55	17
**Kobialka, 2023** **[31]**	Retrospective Cohort study	25	51	12

NA means “not available”.

**Table 2 cancers-16-01929-t002:** Survival outcomes of patients who underwent CRS + HIPEC.

Author	N CRS + HIPEC	Median OS	1 y Rate Survival	3 y Rate Survival	5 y Rate Survival	DFS	30-Day Mortality
**Yang, 2011 [8]**	34	11	41.2	5.9	NA	NA	NA
**Yarema, 2014 [9]**	20	12	68.8	NA	NA	NA	NA
**Wu, 2015 [10]**	11	28.2	79.5	NA	NA	NA	0
**H-T Wu, 2016 [11]**	38	17	71.1	35.8	6.4	NA	NA
**Boerner, 2016 [12]**	50	14.3	58	32	NA	NA	NA
**Chia, 2016 [13]**	81	17.3	NA	NA	18	NA	3
**Topal 2017 [14]**	32	16	71.9	14.1	3.5	7.8	NA
**Caro, 2018 [15]**	35	16	70.8	21.3	21.3	7	NA
**Kim, 2018 [16]**	38	19	NA	NA	NA	NA	2
**Manzanedo, 2019 [17]**	88	21.2	79.9	30.9	27.5	11.6	NA
**Bonnot, 2019 [18]**	180	18.6	67.9	27.1	20.21	11.6	4
**Rau, 2019 [19]**	58	9.8	40.9	12.1	0	NA	NA
**Rau, 2020 [20]**	235	13	NA	NA	NA	NA	12
**Zhong-He Ji, 2020 [21]**	125	10.7	43.8	18.6	15.7	NA	1
**Rosa, 2021 [22]**	23	NA	NA	NA	27	NA	1
**Bagdwell, 2021 [23]**	20	22.1	90	50	28	7	0
**Marano, 2021 [24]**	91	20.2	62	44	20.4	7.3	NA
**Somashekhar, 2022 [25]**	16	17	NA	NA	NA	12	0
**Santullo, 2023 [26]**	20	17	75	14.7	2	12	0
**Buckarma, 2023 [27]**	22	NA	96	78	55	NA	0
**Green, 2023 [28]**	41	24.9	NA	25	NA	7.4	0
**Rau, 2023 [29]**	59	14.9	58.2	13.6	NA	7.1	NA
**Allievi, 2023 [30]**	27	NA	60.3	30.1	NA	NA	NA
**Kobialka, 2023 [31]**	25	16	NA	NA	NA	NA	2

NA means “not available”.

## Data Availability

The data presented in this study are available on request from the corresponding author (accurately indicate status).

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
