# Peer review of "Efficacy of Cytoreductive Surgery (CRS) + HIPEC in Gastric Cancer with Peritoneal Metastasis: Systematic Review and Meta-Analysis"

_cancers, 2024, doi:10.3390/cancers16101929_

Round 1
Reviewer 1 Report
Comments and Suggestions for Authors
Thanks for submitting the systematic review. I do have a few comments and minor suggestions.
Major points.
1. Where do you think the current role of immunotherapy is in these patients? I no longer think systemic chemotherapy alone is the standard of care in metastatic gastric cancer necessarily. Please comment/ discuss.
2. You do mention the Glehen et al publication in your discussion but note was not included in the review itself. Is this due to the exclusion of publications prior to 2011? It does seem to represent one of the largest experiences in this area.
3. In the review, you don't summarize the systemic therapy patients are receiving for the CRS/HIPEC group. Most do not get up-front CRS/HIPEC I presume? But rather get systemic treatment and if doing well/ responding go on to possible CRS/HIPEC? Can you put some data around this in the results and also discuss whether this is a potential selection factor?
4. Do you feel with majority of studies being retrospective and significant heterogeneity (e.g. based on patient selection as well as significant variation in CRS/HIPEC protocols) that the meta-analysis component of your study is valid/ warranted? Would it have been more appropriate to stop at a scoping review?
5. In the discussion, you mention the Italian guidelines recommending systemic treatment and supportive care. However, the newest NCCN guidelines does suggest 'considering' CRS/ HIPEC in well selected patients. I would suggest discussing as well.
6. I would make a couple of suggestions about your language in the discussion. You do comment in the discussion (paragraph 4) that your study 'strengthens the evidence supporting the survival benefits of CRS/ HIPEC.' I would suggest 'summarizes the evidence potentially supporting the survival...'. My suggestion is based on the reality that there are many limitations still (as you point out) based on clearly selected patients and mainly retrospective data. The study doesn't necessarily add to the data out there although agree could add to the literature.
7. Similarly, I am unsure about the language of your conclusion. I think 'could guarantee better survival outcomes compared to systemic chemotherapy' is too strong even with the caveats you mention. Again, given the very small number of patient overall treated in this fashion captured by the literature and the limitations you note, I would suggest something more neutral...'shows promise and may have better survival outcomes in a small subset of patients' or something similar?
Minor edits. 1. In the intro paragraph - I would suggest '4-6 months' rather than '4,6'
2. I would suggest a reference in paragraph 2; even if repeated.
3. In paragraph 3, I would suggest 'controversial' rather than 'controverse'
Comments on the Quality of English LanguageVery good. I suggested a couple minor edits to the authors.
Reviewer 2 Report
Comments and Suggestions for Authors
The authors produced a review, according to PRISMA guidelines, of a relatively rare but important niche in the treatment of advanced gastric cancer: cytoreductive surgery (CRS) plus HIPEC. The analysis is critical, considering inherent weaknesses: retrospective nature of most trials with biases and confounders. The conclusion that CRS + HIPEC results in a better survival than systemic chemotherapy may contribute to the design of clinical trials in subgroups of this malignancy with a bad prognosis.
Reviewer 3 Report
Comments and Suggestions for Authors
Metastatic lesions that are localized, technically resectable at diagnosis, present a certain response to preoperative chemotherapy, and present favorable survival outcomes with local treatments, sometimes in combination with chemotherapy, are recognized as oligometastasis in the field of gastric cancer. Peritoneal dissemination is an indicator of unresectable metastasis; however, patients are potentially eligible for surgical resection if the dissemination in the peritoneal cavity is localized near the stomach, which is classified as P1 disease in the Japanese classification of gastric carcinoma, first edition. The standard treatment for patients with peritoneal metastasis localized near the stomach has not been established. CRS + HIPEC treatment followed by postoperative chemotherapy is one option. So, when gastric cancer patients who can receive this CRS + HIPEC treatment, the patients belong to oligometastatic of gastric cancer, and the prognosis is better than multiple metastases. While, this need more randomized controlled trials.
Comments on the Quality of English LanguageModerate editing of English language required.
Round 2
Reviewer 3 Report
Comments and Suggestions for Authors
I think it is worth publishing in the journal.
Comments on the Quality of English LanguageI think it is worth publishing in the journal.